# Molecular Evolution of Far-Red Light-Acclimated Photosystem II

**DOI:** 10.3390/microorganisms10071270

**Published:** 2022-06-22

**Authors:** Christopher J. Gisriel, Tanai Cardona, Donald A. Bryant, Gary W. Brudvig

**Affiliations:** 1Department of Chemistry, Yale University, New Haven, CT 06520, USA; christopher.gisriel@yale.edu; 2Department of Life Sciences, Imperial College London, London SW7 2AZ, UK; t.cardona@imperial.ac.uk; 3Department of Biochemistry and Molecular Biology, The Pennsylvania State University, University Park, PA 16802, USA; dab14@psu.edu; 4Department of Molecular Biophysics and Biochemistry, Yale University, New Haven, CT 06520, USA

**Keywords:** photosynthesis, cyanobacteria, ancestral sequence reconstruction, chlorophyll *f*, chlorophyll *d*, far-red light photoacclimation, *Synechococcus* sp. PCC 7335

## Abstract

Cyanobacteria are major contributors to global carbon fixation and primarily use visible light (400−700 nm) to drive oxygenic photosynthesis. When shifted into environments where visible light is attenuated, a small, but highly diverse and widespread number of cyanobacteria can express modified pigments and paralogous versions of photosystem subunits and phycobiliproteins that confer far-red light (FRL) absorbance (700−800 nm), a process termed far-red light photoacclimation, or FaRLiP. During FaRLiP, alternate photosystem II (PSII) subunits enable the complex to bind chlorophylls *d* and *f*, which absorb at lower energy than chlorophyll *a* but still support water oxidation. How the FaRLiP response arose remains poorly studied. Here, we report ancestral sequence reconstruction and structure-based molecular evolutionary studies of the FRL-specific subunits of FRL-PSII. We show that the duplications leading to the origin of two PsbA (D1) paralogs required to make chlorophyll *f* and to bind chlorophyll *d* in water-splitting FRL-PSII are likely the first to have occurred prior to the diversification of extant cyanobacteria. These duplications were followed by those leading to alternative PsbC (CP43) and PsbD (D2) subunits, occurring early during the diversification of cyanobacteria, and culminating with those leading to PsbB (CP47) and PsbH paralogs coincident with the radiation of the major groups. We show that the origin of FRL-PSII required the accumulation of a relatively small number of amino acid changes and that the ancestral FRL-PSII likely contained a chlorophyll *d* molecule in the electron transfer chain, two chlorophyll *f* molecules in the antenna subunits at equivalent positions, and three chlorophyll *a* molecules whose site energies were altered. The results suggest a minimal model for engineering far-red light absorbance into plant PSII for biotechnological applications.

## 1. Introduction

Photosystem II (PSII) is the water–plastoquinone photooxidoreductase that is the keystone enzyme of oxygenic photosynthesis [1]. By coupling solar energy to water oxidation, PSII generates the reducing equivalents that support photosynthetic carbon dioxide fixation by plants, algae, and cyanobacteria. These organisms use chlorophyll (Chl) *a,* which absorbs light in the visible region of the solar spectrum from 400 to 700 nm [2], as the primary light-harvesting and photochemically active pigment in PSII. However, by additionally synthesizing and inserting Chls *d* and *f* into far-red light-acclimated PSII (FRL-PSII), some terrestrial cyanobacteria can acclimate to thrive in shaded environments that are naturally enriched in lower energy, far-red light (FRL; λ = 700 to 800 nm) [3,4,5,6]. Because the absorbance spectra of Chls *d* and *f* are red-shifted relative to that of Chl *a* [7,8], these Chls allow lower energy FRL to support water oxidation by FRL-PSII [3,5,9,10]. Understanding the molecular basis of how and where Chls *d* and *f* are incorporated in PSII is of great interest because it could provide design principles that might pave the way for engineering shade tolerance into crops [11,12].

The implementation of Chls *d* and *f* in FRL-PSII by some cyanobacteria is part of a wider acclimation process called far-red light photoacclimation, or FaRLiP [3,9]. FaRLiP-capable cyanobacteria are abundant and quite widespread [9,13,14,15,16,17,18,19], with members occurring in all five major taxonomic sections of the cyanobacteria [20]; thus, they substantively contribute to global carbon fixation. When grown under FRL, cyanobacteria that perform FaRLiP upregulate a FRL-specific cluster of 20 genes that encodes proteins involved in the biosynthesis of Chl *f*, as well as FRL-specific subunits found in peripheral phycobiliprotein antenna complexes and the two FRL-acclimated photosystems [3,13]. In PSII, FRL-specific isoforms of the four core subunits, PsbA (D1), PsbB (CP47), PsbC (CP43), and PsbD (D2), together with a peripheral subunit called PsbH, are incorporated into the FRL-PSII complex rather than isoforms found when the cells are grown in visible light (VL) [4]. For clarity, we hereafter refer to the PSII subunits by the name of their gene product without the designation or allele number. For example, the D1 subunit expressed during growth under VL will be referred to as VL-PsbA, and the corresponding isoform expressed during growth under FRL will be referred to as FRL-PsbA.

To uncover the architectural details of FRL-PSII, the structure of a monomeric FRL-PSII core complex from *Synechococcus* sp. PCC 7335 (hereafter, *Synechococcus* 7335) was recently obtained by cryo-electron microscopy (cryo-EM) [21]. Although the complex lacked the dimeric configuration typically observed in a PSII holocomplex and various peripheral subunits including FRL-PsbH, it retained nearly all Chl binding sites. This allowed for the assignment of four sites that bind Chl *f* molecules in the antenna subunits and one site, Chl_D1_, that binds Chl *d* among the cofactors that comprise the electron transfer chain. It was found that the protein confers binding specificity due to FRL-specific residues that preferentially accommodate the structures of Chls *d* or *f*; these Chls exhibit polar formyl moieties at either the C2 position (Chl *f*) or the C3 position (Chl *d*) on the tetrapyrrole ring, whereas Chl *a* exhibits hydrophobic methyl or vinyl moieties at those locations, respectively. Additionally, the FRL-PsbD and FRL-PsbC subunits contained conserved FRL-specific residues on the stromal surface that are probably complementary to residues of FRL-specific allophycocyanin subunits of the peripheral phycobiliprotein complex [21,22].

Because FRL-PSII is such an important bioenergetic system in the contribution of cyanobacteria to global carbon fixation, understanding its evolution and diversity among extant organisms is also of great interest. Recent evolutionary studies have suggested that FaRLiP has been inherited in cyanobacteria mostly vertically from a common ancestor that likely existed during the early to mid-Proterozoic [14]. Building upon the basis of the structural data previously obtained [21], one can use sequence alignments and phylogenetics to gain detailed molecular insights into how FRL-PSII arose and subsequently evolved and diversified. Here, we have: (a) analyzed phylogenetic trees of PSII subunit sequences to derive the evolutionary events that led to modern FRL-PSII, (b) used all available protein sequences of FRL-PsbA, FRL-PsbB, FRL-PsbC, FRL-PsbD, and FRL-PsbH to reconstruct the ancestral sequence of each, and (c) performed a thorough structural analysis of the *Synechococcus* 7335 FRL-PSII structure with homology models of the ancestral sequences and PSII structures from cyanobacteria incapable of performing FaRLiP. These comparisons reveal insights into how the characteristics of modern FRL-PSII evolved over time and provide new insights into the diversity of FRL-PSII among modern cyanobacteria. The findings also provide a minimal model for FRL-specific characteristics that can be used as a starting point to engineer FRL absorbance into crops.

## 2. Materials and Methods

### 2.1. Phylogenetic Tree Construction

All non-redundant and complete amino acid sequences were downloaded from the National Center for Biotechnology Information (NCBI) RefSeq database using PSI-BLAST with default settings and restricted to the phylum Cyanobacteria. PsbA and PsbD sequences were collected on the 17 August 2021, PsbB and PsbC sequences on the 6 January 2022, and PsbH sequences on the 14 February 2022. Redundancy was decreased to 98% sequence identity using the following tool https://web.expasy.org/decrease_redundancy/ (accessed on the same dates the sequence datasets were collected) from the Swiss Bioinformatics Resource Portal [23,24]. In total, 990 PsbA, 487 PsbB, 436 PsbC not including any “CP43-like” Chl-binding proteins, 274 PsbD, and 464 PsbH sequences were kept for further analysis. Sequence alignments were performed with Clustal Omega [25] using five combined guided trees and hidden Markov model iterations [26].

Maximum likelihood unrooted phylogenies were inferred using IQ-TREE multicore version 2.0.3 [27]. Best fit parameters were calculated using its in-built function ModelFinder [26]. Support values were estimated using ultrafastbootstrap with >1000 iterations until the correlation coefficient of split occurrence frequencies converged [28,29]. The average likelihood ratio test method to estimate branch support values was also used at the same time [30]. Ancestral sequences were calculated during tree inference by activating the function *-asr*. The ancestral sequence reconstruction function of IQ-TREE does not estimate ancestral sequence insertions or deletions; therefore, prior to homology modeling, all insertions were manually removed using the respective sequence from the *Synechococcus* 7335 FRL-PSII as the template. Trees were visualized using the software Dendroscope version 3.8.1 [31]. All sequence alignments, ancestral sequences and trees used in this study are provided in Appendix A.

### 2.2. Homology Modeling

To create homology models of ancestral FRL-PsbA, FRL-PsbB, FRL-PsbC, and FRL-PsbD, the subunit of interest was isolated from the cryo-EM structure of monomeric apo-FRL-PSII from *Synechococcus* 7335 deposited under PDB accession code 7SA3 using PyMOL [32]. For the above-mentioned ancestral sequences, each of these was used as the template to create homology models using the Swiss-Model server [33]. To create the homology model of ancestral FRL-PsbH, the PsbH subunit was isolated from the cryo-EM structure of the PSII holocomplex from *Synechocystis* sp. PCC 6803 (hereafter, *Synechocystis* 6803) [34], PDB accession code 7N8O.

## 3. Results

### 3.1. Phylogenetic Analysis

To provide insight into the evolution of the FRL-PSII subunits, we generated phylogenetic trees for each of the core subunits: PsbA, PsbD, PsbC, PsbB, and PsbH (Figure 1 and Appendix A). The evolution of PsbA has been studied in detail [35,36,37,38] and the tree shown in Figure 1 is generally consistent with those shown before. The tree includes two PsbA paralogs of importance for FaRLiP: (1) the Chl *f* synthase, also known as ChlF [39,40,41] or Group 1 (G1); and (2) FRL-PsbA, which enables water oxidation under FaRLiP. PsbA evolution is characterized and dominated by continuous gene duplications leading to the extensive diversity of paralogs in extant organisms. Some of these duplications are ancient and are suggested to have occurred before the most recent common ancestor (MRCA) of cyanobacteria [36,37], which is likely to have had multiple copies of PsbA, similar to extant cyanobacteria [42]. The earliest duplications are those that led to the emergence of the atypical PsbA variants (G0 to G2 sequences) and the so-called microoxic forms, Group 3. The PsbA tree calculated in this study, as well as that recently published by Sheridan et al. [35] shows that the duplication leading to FRL-PsbA is also likely to have occurred before the MRCA of cyanobacteria, prior to a duplication leading to another subgroup of sequences that was denoted in Sheridan et al. as D1^INT^ [35]. 

The other subunits of PSII have not left a record of gene duplications antedating the last common ancestor of cyanobacteria, with the potential exception of the duplication leading to the family of Chl-binding proteins derived from PsbC (e.g., IsiA), which are unable to support water oxidation or replace PsbC in active PSII complexes [38]. Therefore, the duplications leading to FRL-PsbB, FRL-PsbC, FRL-PsbD, and FRL-PsbH are likely to have occurred after the MRCA of cyanobacteria, and consequently, after that leading to FRL-PsbA. FRL-PsbC and FRL-PsbD have a basal branching position. In contrast, the duplications leading to FRL-PsbB and PsbH appear to have occurred later, at a time mostly coincident with the radiation of cyanobacteria leading to most major groups [43].

The trees in Figure 1 and Appendix A also show that the different FRL subunits feature different rates of sequence change when compared with each other and within their respective groups, with FRL-PsbH and FRL-PsbC showing particularly longer branches. It is also worth noting that there is some variation in the length of the branch leading to the MRCA of each FRL subunit group, marked with an orange circle in Figure 1. This suggests that before the diversification of the FaRLiP gene cluster, each FRL subunit ancestor had accumulated a different number of FRL-specific substitutions since the point of duplication, with the FRL-PsbH ancestor having accumulated the most changes per amino acid position compared with the VL form, followed by the FRL-PsbC, FRL-PsbA, FRL-PsbB, and FRL-PsbD ancestors. We note that none of the trees showed evidence that the FRL sequences had any close relationship to those in the Chl *d*-producing *Acaryochloris* sp. strains. This observation supports the hypothesis that the facultative FaRLiP response is unrelated to the constitutive expression of Chl *d* exhibited by *Acaryochloris* strains, and that the two strategies probably evolved convergently.

### 3.2. Sequence Annotations and Structural View

To explore the predicted sequence for each ancestor of the FRL-specific subunits, and to estimate the similarity of these subunits among species, we reconstructed the ancestral sequence for each (orange circle in Figure 1). We then compared these ancestral sequences with a subset of sequences from extant organisms to identify residues of interest more conveniently (Appendix A). The subset of sequences comprises the FRL and VL sequences from representative cyanobacteria of different taxonomic sections including *Aphanocapsa* sp. GSE-SYN-MK-11-07L (*Aphanocapsa* GSE), *Pleurocapsa* sp. PCC 7327 (*Pleurocapsa* 7327), *Synechococcus* 7335, and *Fischerella thermalis* PCC 7521 (*Fischerella* 7521). These were realigned using Clustal Omega [25]. We annotated those residues that were conserved among the FRL sequences from the extant cyanobacteria but were also dissimilar from the corresponding VL sequences (green highlights in Appendix A), and we annotated those residues from the ancestral sequence that were the same as the FRL-specific residues in extant cyanobacteria (yellow highlights in Appendix A).

From these annotations (Appendix A), we also calculated the percentage of residues in the FRL-specific subunits from *Synechococcus* 7335 that appear to be FRL-specific in the subset and are conserved with the corresponding ancestral sequence, and those that are not (Table 1). A relatively small fraction of the total residues in those subunits, only 74 of 1768 (4.2%), appear to be FRL specific. Most of these, 53 of the 74, are also conserved in the ancestral sequences. Of the four core subunits, three have FRL-specific residues that are all conserved in their corresponding ancestral sequences: FRL-PsbA, FRL-PsbC, and FRL-PsbD. This implies that these subunits have changed little in the history of FRL-PSII. The FRL-PsbD subunit has the smallest fraction of residues that appear FRL-specific, only 1.7%, which is consistent with it not binding Chls *d* or *f* [21]. It is also consistent with the short branch that separates the group in the phylogeny from all other PsbD sequences (Figure 1). FRL-PsbA and FRL-PsbC have 3.3% and 3.9% of their residues that appear FRL specific, respectively, and each binds a single FRL-absorbing Chl. FRL-PsbB has more FRL-specific residues than the other three core subunits, 6.3%, but only approximately one-third of those residues are conserved in the ancestral sequence. This indicates that FRL-PsbB presents greater lineage-specific differences than the other FRL-specific core subunits, despite this not being immediately obvious by inspecting the phylogenies alone. Finally, the FRL-PsbH is the only non-core subunit with a FRL isoform. It contains five FRL-specific residues that are all close to the N-terminus, and four of these are conserved in the ancestral sequence. This might indicate that FRL-PsbH associates with the FRL-allophycocyanin antenna. Generally, these observations suggest that the FRL-specific PSII subunits are under selective pressure to maintain a small number of specific sites that are important for FRL-based light absorption and photochemistry, but some extant FRL-PsbB sequences may contain some features that were not present in the ancestor of FRL-PSII.

To visualize the number and frequency of the FRL-specific residues from a structural perspective, we mapped those sites to the FRL-specific subunits from the cryo-EM structure of monomeric apo-FRL-PSII from *Synechococcus* 7335 [21], which contained a FRL-PsbH homology model (Figure 2) occupying the usual position based on the cryo-EM structure of the PSII holocomplex from *Synechocystis* 6803 [34] (Appendix A). Consistent with the sequence alignment analysis (Appendix A), residues are frequently observed that are FRL-specific and are conserved in the ancestral sequences (yellow spheres in Figure 2). Residues that are FRL-specific but are not conserved in the ancestral sequence are mostly found in FRL-PsbB and one in FRL-PsbH (green spheres in Figure 2). Notable regions of FRL-specific residues that are maintained in the ancestral sequences are at the stromal surface of PsbC and PsbD, the cluster of conserved residues near the Chl *d* found in the electron transfer chain, and the N-terminal region of FRL-PsbH (Figure 2).

### 3.3. Conserved Features of FRL-PsbA

All 12 of the conserved FRL-specific residues in the subset of FRL-PsbA sequences from extant cyanobacteria (Appendix A) were estimated in the ancestor with posterior probabilities (PP) above 0.99. All form a cluster nearby the formyl moiety of the Chl *d* in the Chl_D1_ site, which is consistent with previous observations [21] (Figure 2 and Figure 3). This implies that the Chl *d* in the Chl_D1_ site was present in the ancestor to extant FRL-PSII complexes, and that there is strong selective pressure to maintain those nearby FRL-specific residues. This cluster of conserved ancestral FRL-specific residues is found within an ~15 Å sphere between the formyl moiety of the Chl *d* and PsbI, adjacent to the PSII dimerization interface (Figure 3). Some notable residues are those that are found in an H-bonding network with the formyl moiety of Chl *d*, PsbA3-Thr155 and Tyr120, as reported previously [21]; however, other nearby residues are likely to participate in stabilizing the FRL-specific interactions or to facilitate Chl *d* insertion/binding during PSII assembly.

### 3.4. Conserved Features of FRL-PsbD

All FRL-specific residues in the subset of FRL-PsbD sequences from extant organisms (Appendix A) are conserved in the ancestral sequence with high confidence (PP ≥ 0.91). Two of the conserved residues are nearby the Chl *a* molecule in the P_D2_ site of the electron transfer chain (Appendix A). As reported previously [21], the hydroxyl group of the PsbD3-Tyr191 sidechain donates an H-bond to the keto-oxygen atom of the 13^2^ methoxycarbonyl moiety of P_D2_. Interestingly, despite this being confidently predicted in the ancestor, out of the 37 FRL sequences in this dataset, there are six instances of independent reversal of the PsbD3-Tyr191 to Phe (Appendix A), which could indicate that (a) the H-bond may not be strictly necessary for FRL function or (b) a different kind of H-bonding interaction is present in the organisms that have Phe at that position. Adjacent to PsbD3-Tyr191 is another conserved FRL-specific residue that is present in the FRL-ASR and strictly conserved in all FRL sequences, PsbD3-Met286. The position of the Met sidechain is shifted slightly away from position 191 (Appendix A), allowing enough space for the H-bonding interaction while maintaining a hydrophobic environment observed in the VL sequences (either Ile or Val, Appendix A). The strict conservation of this Met residue is consistent with the hypothesis that the six FRL-PsbD sequences that lack the H-bonding PsbD3-Tyr131 maintain some other kind of H-bonding interaction, perhaps with a water molecule.

Another region of FRL-PsbD with FRL-specific characteristics is the stromal surface loop between transmembrane helices 4 and 5. This loop is present in all cyanobacterial PSII structures and is unique in that it stretches over the D1 subunit and interacts with PsbC (Figure 4). In this loop, two residues are found in the subset of FRL-PsbD sequences and are also predicted in the FRL ancestor: Gly234 (PP = 0.99) and Ser236 (PP = 0.91) (Figure 4A,B). This is consistent with the suggestion that FRL-allophycocyanin binds to the stromal surface of FRL-PSII in extant FaRLiP-capable cyanobacteria, to the FRL-PsbC and FRL-PsbD subunits [4,21,22,44], possibly to stabilize the interaction and/or tune energy transfer to the core. It also suggests that PsbD-Ser228 and Gly234 were present and performing a similar function in the MRCA of FRL-PSII complexes well over a billion years ago. However, these two residues are not strictly conserved in all extant FRL sequences: Gly234 and Ser236 are found in 27 and 25 out of the 37 sequences, respectively, which suggests that the sites may be under weaker selective pressure compared to strictly conserved sites. Note that these FRL-specific surface residues are close to the Chl *f* in PsbC and a Chl *a* molecule in PsbC that exhibits a FRL-specific H-bond, as described below.

### 3.5. Conserved Features of FRL-PsbC

As was the case for FRL-PsbA and FRL-PsbD, all FRL-specific residues in the subset of FRL-PsbC sequences from extant cyanobacteria are conserved in the FRL-PsbC ancestral sequence (Appendix A). FRL-PsbC contains one Chl *f* molecule that was suggested to serve as a bridging Chl for energy transfer from FRL-allophycocyanin to the electron transfer chain cofactors [21]. Similar to FRL-PsbD, there are FRL-specific residues on the stromal surface of FRL-PsbC that are found in the ancestral sequence (Figure 4). There is also a short deletion (dashed line in Figure 4) that corresponds to the FRL-PsbC sequence that is conserved with the FRL-PsbC ancestral sequence. Along with the conserved residues from the FRL-PsbD loop (Figure 4A), these observations are consistent with the idea that an ancestral FRL-allophycocyanin protein complex bound to the stromal surface of the FRL-PSII ancestor near the FRL-PsbC side of the complex similar to how this occurs in extant FRL-PSII complexes. Obtaining structural data for FRL-allophycocyanin in complex with PSII may aid in determining the roles of the individual conserved residues at their interface surfaces.

An additional region with multiple FRL-specific residues in FRL-PsbC occurs in its fifth transmembrane helix. Here, four FRL-specific residues are strictly conserved and are predicted in the FRL-PsbC ancestral sequence (Figure 5) with high confidence (PP > 0.99). The backbone amide nitrogen atom of FRL-PsbC-Asn281 donates an H-bond to a water molecule that is, in turn, an H-bond donor to the formyl moiety of the single Chl *f* molecule in FRL-PsbC, Chl *f* 507 as reported previously [21]. Interestingly, there is also a FRL-specific interaction with an adjacent Chl *a* molecule, Chl *a* 505, which is ~5.1 Å from Chl *f* 507 (edge-to-edge). Specifically, the nitrogen atom of the sidechain for FRL-PsbC-Gln277 donates an H-bond to the 13^1^-keto oxygen atom of Chl *a* 505 (Figure 5). H-bonding to this oxygen atom of Chl *a* is well-characterized to result in a bathochromic shift of the absorbance spectrum due to a decrease in site energy [45], which is likely also the case here, and is probably important for energy transfer to the core along with the presence of Chl *f* 507. In addition, FRL-PsbC-Gly284 is conserved in the extant and ancestral FRL sequences, but is instead a bulkier, hydrophobic Leu in the VL sequences. In *T. vulcanus* and *Synechocystis* 6803, this position is Met and Leu, respectively [34,46]. The extra volume available in the FRL-PSII structure allows Chl *a* 505 to be shifted slightly away from the Chl in position 507, allowing space for the water molecule that is important for the latter site to bind Chl *f*. This water is not found in the PSII structures from *T. vulcanus* or *Synechocystis* 6803. These observations suggest that the Chl *f* in site 507, the H-bond to the 13^1^-keto oxygen atom of Chl *a* 505, and the shift of Chl *a* 505 were all present in the FRL-PSII ancestor.

Another interesting observation regarding FRL-PsbC is the absence of three residues found in all VL-PsbC, with the exception of the basal *Gloeobacterales* clade. These are PsbC-Glu142, Tyr143, and Ser144 in the structure of *T. vulcanus*, located in the stromal surface between the second and third transmembrane helices. The presence of this unique gap does not appear to have any relationship to FaRLiP, but it is consistent with the phylogenetic placement of the FRL group indicating an origin from an early duplication event prior to the main cyanobacterial radiation.

### 3.6. Conserved Features of FRL-PsbB

Of the core subunits, FRL-PsbB has the largest percentage of FRL-specific residues conserved among the extant cyanobacteria in the subset, but not necessarily in the ancestral sequence (Appendix A). This suggests that unlike the other core subunits, FRL-PsbB proteins have acquired more lineage-specific changes since their most recent common ancestor. Interestingly, only one of the Chl *f* sites in FRL-PsbB, Chl *f* 614, appears to be conserved based on the sequence alignments (Figure 6). Whereas all the FRL-specific residues near Chl *f* 614 are conserved in the FRL-PsbB ancestral sequence (PP > 0.99), the FRL-specific residues near Chl *f* 605 and Chl *f* 608 are not, including those near their C2 formyl moieties that confer binding specificity for Chl *f* over Chl *a*. This suggests that the FRL-PSII ancestor did not bind Chl *f* molecules at sites 605 and 608, and some extant representatives also might not. For example, the 18 FRL sequences in our dataset cluster in three distinct clades in the maximum likelihood phylogeny (Appendix A). Out of these sequences, only six have PsbB-Trp268 needed in the H-bonding network to Chl *f* 608 (Figure 6C), grouping together as a monophyletic clade. This subgroup coincidentally includes, besides *Synechococcus* 7335, *Aphanocapsa* GSE, and *Pleurocapsa* 7327, the sequences from the heterocystous *Fischerella* spp. and those from other very closely related strains with over 98% sequence identity (e.g., *Chlorogloeopsis*). The *Fischerella* spp. group is separate from other heterocyst-forming cyanobacteria (e.g., *Calothrix, Rivularia*, etc.), which instead maintain Phe in this position as found in the ancestral FRL-PsbB sequence as well as VL-PsbB sequences and the PsbB sequences from the FaRLiP-incapable *T. vulcanus* and *Synechocystis* 6803. The same is true for positions Phe33 and Leu62 near Chl *f* 605. It is plausible that this may represent a case of gene replacement via horizontal gene transfer, although convergent evolution cannot be entirely ruled out. It suggests that Chls *f* 605 and 608 may be unique to the subgroup containing *Synechococcus* 7335, which highlights the need for more structural data on FRL-PSII from other cyanobacterial species.

Another important observation occurs at position PsbB-Val244 in the sequence of *Synechococcus* 7335, which is found as Thr244 in 16 out of the 18 FRL sequences, and as Ala in VL-PsbB sequences (and also the PsbB sequences from the FaRLiP-incapable *Synechocystis* 6803 and *T. vulcanus*). The two sequences in the FRL dataset without Thr244 are that of *Synechococcus* 7335, from which the apo-FRL-PSII cryo-EM structure was solved, and one from a strain of *Nodosilinia* also with Val at the same position. Based on homology modeling, the sidechain of PsbB-Thr244 was previously suggested to confer Chl *f*-binding in site 611 of FRL-PSII from *Fischerella* 7521 by donating an H-bond to a C2 formyl moiety [47]. This assignment was also supported by the lack of an axial-coordinating His sidechain for this site, which is more frequently observed for Chl *f*-containing sites than it is for Chl *a*-containing sites [48,49]. The ancestral sequence predicts a Thr in position 244, so if Chl *f* occupies this site in many extant FRL-PSII complexes, it is also likely to have been found in the FRL-PSII ancestor. To investigate this further, we modeled a Chl *f* into the Chl 611 site from the *Synechococcus* 7335 apo-FRL-PSII structure and modeled a Thr in position 244 instead of Val, occupying the identical rotamer position. Indeed, this approximation places the hydroxyl moiety of Thr244 within H-bonding distance of the C2 formyl moiety of Chl *f* 611 (Appendix A). Additionally, the Chl in site 611 is related by pseudo-*C*_2_ symmetry to the Chl *f* found in FRL-PsbC. Such symmetry in antenna Chl *f* sites was also observed in FRL-acclimated photosystem I structures [50], further supporting Chl site 611 binding Chl *f* in organisms where FRL-PsbB contains Thr at position 244. Thus, we propose that the Chl 611 site in most FRL-PSII complexes binds Chl *f*, and that this was a characteristic of the ancestral FRL-PSII complex.

### 3.7. Conserved Features of FRL-PsbH

PsbH is known to be involved in PSII assembly by binding to PsbB prior to insertion into the PsbA/PsbD/cytochrome *b*_559_ complex [51]. It is also involved in PSII repair, assisting the incorporation of newly synthesized PsbA into the complex and the binding of bicarbonate [51,52]. Unfortunately, FRL-PsbH is the only FRL-specific subunit in the FRL-acclimated photosystem complexes not yet resolved structurally, as it was not present in the apo-FRL-PSII structure from *Synechococcus* 7335 [21]. Despite its different sequence relative to VL-PsbH isoforms, FRL-PsbH probably binds to the same site relative to the core as is observed in PSII structures from non-FaRLiP strains because only a small sequence region at the N-terminus appears to be FRL specific (Figure 7).

In all PSII structures containing PsbH, the N-terminal region of PsbH extends across the stromal surface near the edge of PsbB where the dimerization interface is found. In the FRL subunit isoforms, including the ancestral sequence, the N-terminus is extended even further, by six or seven residues, and four FRL-specific residues are found that are conserved between FRL-PsbH of extant cyanobacteria and the FRL-PsbH ancestral sequence. This N-terminal region of PsbH surrounds a stromal side Chl *a* molecule in VL-PSII structures, Chl *a* 617, which suggests that FRL-PsbH may alter the site energy of this pigment relative to VL-PsbH. In PSII structures from cyanobacteria incapable of FaRLiP, an H-bond is donated from PsbH-Thr5, which results in red-shifted fluorescence emission from Chl *a* 617 [53], and this residue appears to be conserved in the VL-PsbH sequences from our sequence alignments (Figure 7). In the FRL-PsbH sequences from the subset of extant cyanobacteria and the FRL-PsbH ancestral sequence, a well conserved Ala is found at this position (Appendix A). This suggests that the site energy of Chl *a* 617 is altered in FRL-PSII. In addition to this Ala residue, the three other FRL-specific residues are also close to this site, and all are also present in the FRL-PsbH ancestral sequence with high confidence. Again, their presence suggests that the site energy of Chl *a* 617 is altered in FRL-PSII compared to VL-PSII. It seems unlikely, however, that FRL-PsbH could confer Chl *f*-binding at site 617 because the C2 moiety is positioned away from the stromal surface (i.e., toward the hydrophobic interior), distant from the conserved FRL-specific FRL-PsbH residues. We note that the single Chl *f* site found in PsbB of the *Synechococcus* 7335 apo-FRL-PSII structure, which is conserved in the FRL-PSII ancestor (site 614), and the site that probably contains Chl *f* in many extant species and may have been present in the FRL-PSII ancestor (site 611) are also located close to the stromal side of the complex (Figure 7). These sites are ~20–30 Å from the N-terminal extension of FRL-PsbH, whereas the other two FRL-PsbB Chl *f* sites that were probably not present in the FRL-PSII ancestor are on the lumenal side of the complex (Figure 6). Thus, if the site energy of Chl *a* 617 is altered in FRL, whether in extant or ancestral FRL-PSII complexes, it could be associated with energy transfer with the stromal Chl *f* molecules.

Based on the known function of PsbH in non-FaRLiP cyanobacterial strains [51,52] and its N-terminal alterations in FRL isoforms (Figure 7), FRL-PsbH also seems likely to be involved in PSII assembly and repair and/or dimerization. In non-FaRLiP PSII holocomplex structures, a sulfoquinovosyl-diacylglycerol (SQD) is found in the dimerization interface, very near the N-terminal region of PsbH (Figure 7). The homology models cannot capture the positions of most of the N-terminal residues that comprise the FRL-specific extension of FRL-PsbH or that of its ancestral sequence, and even the residues that can be modeled in the N-terminal regions are likely positioned with low confidence, but the FRL-specific N-terminal extensions could alter the region of the SQD and/or the other interactions at the dimerization interface. Furthermore, it was surprising that the apo-FRL-PSII structure lacked FRL-PsbH, as PsbH is found in all other cryo-EM structures of PSII, even assembly intermediate-like-, apo-, and monomeric-PSII complexes [54,55,56,57]. These observations suggest that FRL-PsbH binds less tightly to the FRL-PSII complex relative to VL-PsbH, which could have implications for details of assembly and repair.

Another unique observation of FRL-PsbH is that out of the 31 sequences in our dataset, 6 contain a second transmembrane helix at the C-terminus not found in the other sequences (Appendix A). These include the sequences from *Aphanocapsa* GSE, strains designated as *Leptolyngbya* and *Elainella*, and some unnamed strains. These sequences make up the basal clades of the FRL-PsbH sequences and therefore indicate that this second helix of FRL-PsbH was likely present in the FRL-PSII ancestor. Further inspection of the sequence revealed that the additional helix has homology to another small subunit found in some, but not all, FaRLiP-capable cyanobacteria, including *Halomicronema hongdecloris* (XM38_010780) and some strains of *Nodosilinea* spp (e.g. WP_194026004.1), *Chroococcidiopsis* spp. (WP_199197231.1) and a few more. This is a 48-residue small subunit annotated as hypothetical, but it is also found within the FaRLiP gene cluster downstream of the *psbH2* gene, suggesting that this could be a novel small subunit of FRL-PSII originating from fission of the ancestral gene for the double helical form of FRL-PsbH.

## 4. Discussion

The phylogenetic analysis presented here suggests several possible transitions prior to the emergence of the MRCA of FaRLiP-capable cyanobacteria and the FaRLiP gene cluster (Figure 8). Taken at face value, the oldest duplication recorded within these PSII subunits is the one leading to ChlF, enabling the synthesis of Chl *f*. Previous work had suggested that ChlF evolved due to a *psbA* gene duplication that occurred before extant cyanobacteria began to diversify [37,38]. However, it is unclear at what point during the evolution of cyanobacteria this paralog evolved to allow Chl *f* synthesis before the MRCA of the FaRLiP-capable cyanobacteria. It is reasonable to assume that the ability to produce and bind, even with low specificity, a few red-shifted pigments alone may have been sufficient to allow survival in FRL-enriched environments prior to the duplications leading to the FRL-specific subunits that could use these pigments more optimally. The second oldest FRL-PSII subunit, as seen in the phylogeny, is FRL-PsbA, which, like ChlF, may have originated from a duplication occurring before the radiation of cyanobacteria. Based on these observations it could be hypothesized that the availability of Chl *f* pigments and the capacity to bind Chl *d* at position Chl_D1_ were some of the earliest innovations towards adapting oxygenic photosynthesis to lower-energy photons, beyond the visible range. It can also be hypothesized that some of the earliest cyanobacteria could have had some ability to use FRL prior to the emergence of the gene cluster as found today. Therefore, it could be predicted that within the more basal clades, which remain poorly sampled, novel FRL use strategies may still be found. It is worth noting that the Chl *d* synthase has not yet been identified [6,58]. While no evidence was found that the FRL sequences had any relationship to those used by *Acaryochloris* spp., there is still a possibility that the adaptations could be still related through the synthesis of Chl *d*. Once the protein(s) responsible for Chl *d* synthesis is(are) found in FaRLiP strains and *Acaryochloris* spp. it may be possible to resolve whether the two adaptations represent separate but convergent evolutionary processes for the use of FRL, or whether the *Acaryochloris* strategy and FaRLiP have a common root that followed diverging paths towards a similar adaptation.

Our analysis suggests that only after the early PsbA duplications, PsbC and PsbD paralogs were added to the emerging FRL-PSII. It is plausible that *psbC* and *psbD* duplicated at the same time since these two genes overlap in most cyanobacteria and in all plastids, including the basal branches, although the overlap is not conserved in the paralogous copies found in the FaRLiP cluster and there is also a stand-alone version of VL-*psbD* outside the FaRLiP cluster [38,59]. The ancestral FRL-PsbD contained a Tyr sidechain that provided a red-shifting H-bond to the 13^2^ methoxycarbonyl moiety of P_D2_ analogous to PsbD-Tyr191 observed in the *Synechococcus* 7335 apo-FRL-PSII cryo-EM structure [21]. The ancestral FRL-PsbD also exhibited some residues on the stromal surface that were important for the binding of or energy transfer from FRL-allophycocyanin. Stromal residues in the ancestral FRL-PsbC served similar roles. The conservation of the stromal residues of these two subunits in the ancestral subunits implies that FRL-allophycocyanin emerged with or before the emergence of FRL-PsbC and FRL-PsbD. If FRL-allophycocyanin evolved first, FRL-PSII may have evolved to optimize energy transfer to it. In any case, the FRL-PsbC ancestor bound a single Chl *f* molecule in site 507 and exhibited a red-shifted site energy for the Chl *a* in site 505. The last stage in the evolution of FRL-PSII was the addition of FRL-PsbB and FRL-PsbH, which may have been added at a later stage to extend FRL-PSII absorption further into the red and to enhance assembly. The FRL-PsbB ancestor contained two Chl *f*-binding sites, 614 and 611, the latter of which is related to the Chl *f* molecule in FRL-PsbC by symmetry with the core. The ancestor of FRL-PsbH appears to have played a role in altering the site energy of a Chl *a* molecule in PsbB, that in site 617, and may have influenced assembly dynamics of the ancestral FRL-PSII complex. We note that every antenna-related characteristic of the FRL-PSII ancestor we have described is present on the stromal side of the FRL-PSII complex (Figure 8). This may support the hypothesis that FRL-allophycocyanin evolved prior to FRL-PSII. 

Most of this description of ancestral FRL-PSII characteristics is maintained in FRL-PSII complexes from extant cyanobacteria, with only the exception of site 611 probably binding Chl *a* and Chl *f* variably among different extant cyanobacteria. This suggests that a minimal model for FRL-PSII may include one Chl *d* (in the Chl_D1_ site of the electron transfer chain), one Chl *f* in each antenna subunit, and possibly three Chl *a* sites (one in each antenna subunit and one at the P_D2_ site in the electron transfer chain) that are altered in their site energy. These modifications are achieved by altering a relatively small percentage of the total residues in the entire complex. These observations will be useful for engineering FRL absorption into crops with the aim of enhancing biomass production in the future.

## Figures and Tables

**Figure 1 microorganisms-10-01270-f001:**
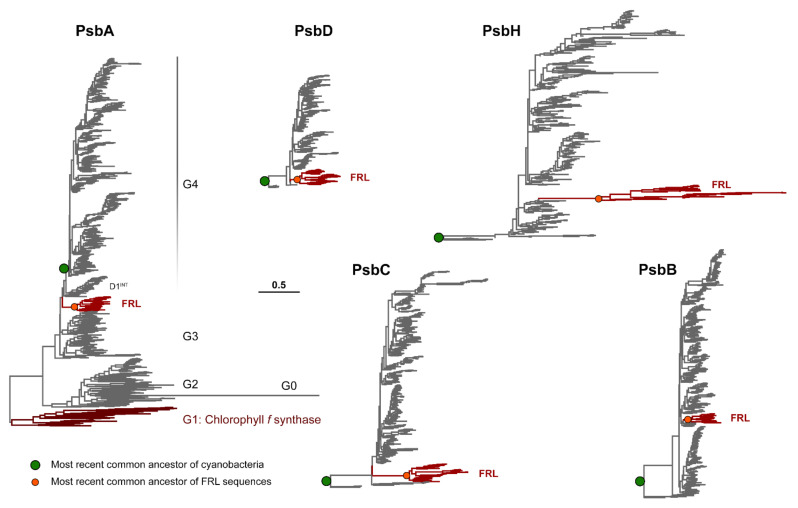
Maximum likelihood phylogenies of FRL-PSII subunits. The PsbA phylogeny has been rooted at the point of divergence of Group 1 sequences as categorized by Cardona et al. [37]. Together with Group 0 and Group 2, these make the “atypical” forms of PsbA characterized by the erosion of the ligand sphere of the water oxidizing cluster. Group 3 and Group 4 include all sequences with a conserved ligand sphere and that can support water oxidation function. In this phylogeny, Group 3 has been split into several subgroups, while the G0 sequences are artificially clustered within Group 2. The MRCA of cyanobacteria (green circle) is defined in this instance as the point in Group 4 PsbA, which contains all standard PsbAs, including those found in the basal genera *Gloeobacter* and close relatives (*Anthocerotibacter/Aurora*): order *Gloeobacterales.* For all other subunits, the trees have been rooted at the branching point of *Gloeobacterales*, and therefore the MRCA of cyanobacteria (green circle) is defined as the last shared ancestor between this and all other lineages. The MRCA of all FRL sequences is marked with an orange circle and it is defined as the last shared ancestor of all extant FRL sequences.

**Figure 2 microorganisms-10-01270-f002:**
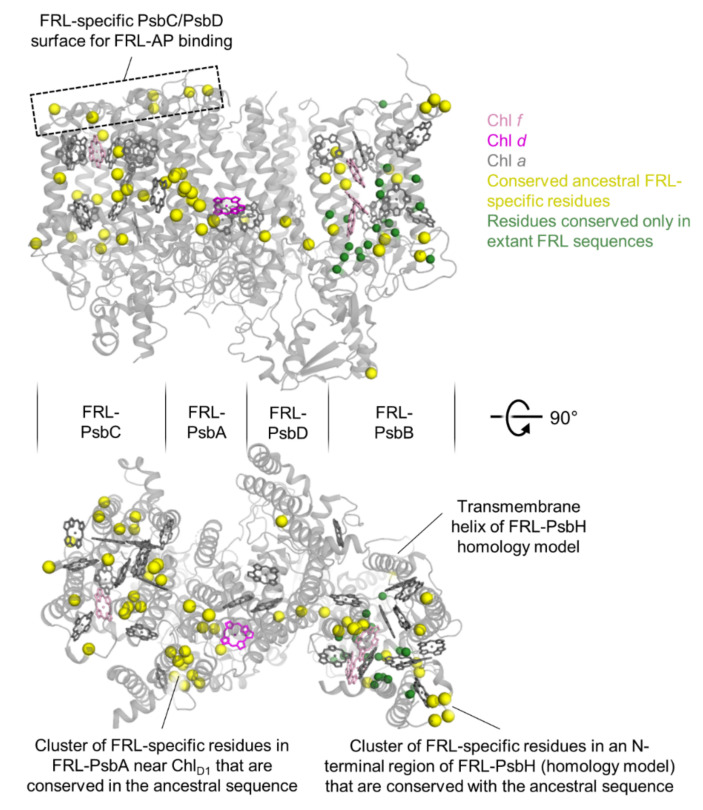
Structural view of conserved FRL-specific residues. The FRL-specific subunits of the apo-FRL-PSII cryo-EM structure from *Synechococcus* 7335 are shown (PDB 7SA3), together with a FRL-PsbH homology model, which was missing in that cryo-EM structure. The tetrapyrrole rings of each Chl molecule from the cryo-EM structure are shown, coloring the Chl *d* molecule in magenta, the Chl *f* molecules in pink, and the Chl *a* molecules in grey. Residues that appear FRL-specific between extant cyanobacteria and are also conserved in the FRL-ancestral sequences are shown as large yellow spheres. Residues that appear FRL-specific between extant cyanobacteria but are not conserved in the FRL-ancestral sequences are shown as smaller green spheres. The approximate regions of the four major core subunits of the two views are denoted by the middle sectioning boundaries. Note that the coloring of this figure corresponds to the highlights in Appendix A.

**Figure 3 microorganisms-10-01270-f003:**
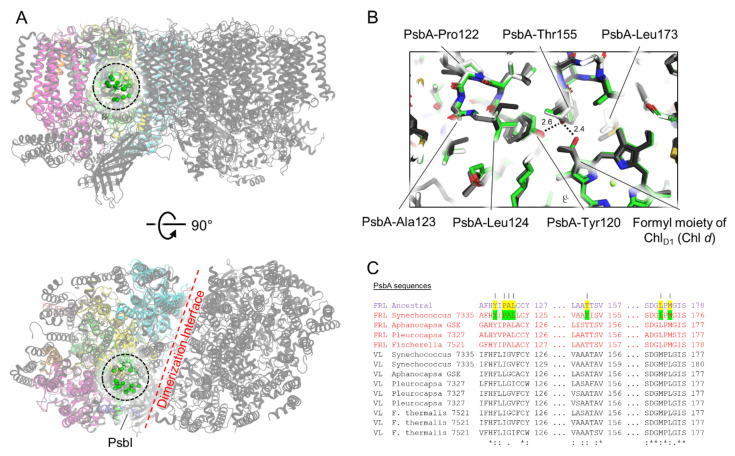
Conserved ancestral FRL-PsbA residues near the Chl_D1_ site of the electron transfer chain. In (**A**), the structure of apo-FRL-PSII from *Synechococcus* 7335 (colors) is shown superimposed with the structure of the PSII holocomplex from *Synechocystis* 6803 (grey, PDB 7N8O). The C_α_ atom from each of the conserved residues in the cluster (black dashed line) near the Chl_D1_ site is shown as a green sphere. (**B**) A magnified view of this region. The *Synechococcus* 7335 apo-FRL-PSII structure (green), the homology model of the FRL-ancestral sequence (white), and two non-FaRLiP holocomplex PSII structures (light and dark grey from *T. vulcanus* [PDB 3WU2] and *Synechocystis* 6803, respectively) are superimposed. H-bonding interactions involving the C3 formyl moiety of Chl *d* in dashed lines with distances in units of Å are also shown. In (**C**), partial sequence alignments are shown that include the FRL-PsbA ancestral sequence, and FRL- and VL-specific PsbA sequences from extant FaRLiP-capable cyanobacteria. FRL-specific residues conserved in extant cyanobacteria are highlighted green in the sequence from *Synechococcus* 7335. If the same position is conserved in the FRL ancestral sequence, it is highlighted yellow. Vertical lines above residue positions in (**C**) correspond to amino acids from the *Synechococcus* 7335 apo-FRL-PSII structure labeled in (**B**). The Clustal Omega sequence conservation identifiers are shown below the alignment.

**Figure 4 microorganisms-10-01270-f004:**
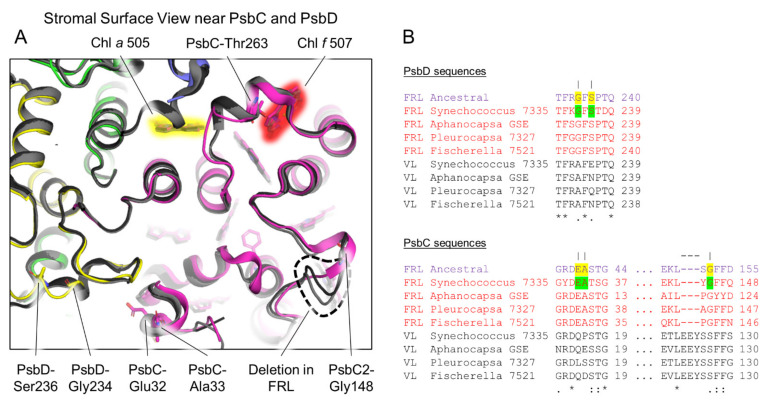
Conservation of stromal surface residues in FRL-PsbC and FRL-PsbD. In (**A**), the *Synechococcus* 7335 apo-FRL-PSII structure (colored) and two non-FaRLiP holocomplex PSII structures (light and dark grey from *T. vulcanus* and *Synechocystis* 6803, respectively) are superimposed. Conserved ancestral residues are shown in stick representation from the *Synechococcus* 7335 apo-FRL-PSII structure only. A PsbC loop absent in FRL-PSII is denoted with a black dashed line. Chl tetrapyrrole rings are shown highlighted in red or yellow. These correspond to the Chl *f* in PsbC and a Chl *a* that exhibits a FRL-specific H-bonding interaction in PsbC, respectively. In (**B**), partial sequence alignments are shown of PsbC and PsbD that include FRL ancestral sequences and FRL- and VL-specific sequences from extant FaRLiP-capable cyanobacteria. Conserved FRL-specific residues in extant cyanobacteria are highlighted green in the sequence from *Synechococcus* 7335. If the same position is conserved in the FRL ancestral sequence, it is highlighted yellow. Vertical lines above residue positions in (**B**) correspond to amino acids labeled in (**A**). The dashed line above the alignment corresponds to the deletion labeled in (**A**). The Clustal Omega sequence conservation identifiers are shown below the alignment.

**Figure 5 microorganisms-10-01270-f005:**
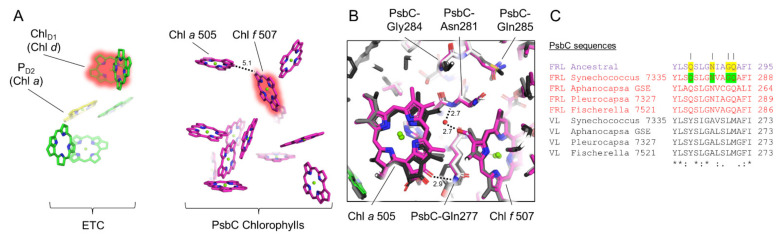
Conservation of FRL-PsbC interactions near Chl *f* 507 and Chl *a* 505. In (**A**), the Chls and pheophytins in the electron transfer chain and FRL-PsbC of the *Synechococcus* 7335 apo-FRL-PSII structure are shown as tetrapyrrole rings only from a stromal-side perspective. The Chls *d* and *f* are highlighted in red, and Chl *a* molecules with FRL-specific H-bonds are additionally labeled. In (**B**), the *Synechococcus* 7335 apo-FRL-PSII structure (magenta), the homology model of the FRL-PsbC ancestral sequence (white), and two non-FaRLiP holocomplex PSII structures (light and dark grey from *T. vulcanus* and *Synechocystis* 6803, respectively) are superimposed. Important H-bonding interactions with distances in units of Å are also shown. In (**C**), a partial sequence alignment is shown that includes the FRL-PsbC ancestral sequence, and FRL- and VL-specific sequences from extant FaRLiP-capable cyanobacteria. Conserved FRL-specific residues in extant cyanobacteria are highlighted in green in the sequence from *Synechococcus* 7335. If the same position is conserved in the FRL-PsbC ancestral sequence, it is highlighted in yellow. Vertical lines above residue positions in (**C**) correspond to amino acids labeled in (**B**). The Clustal Omega sequence conservation identifiers are shown below the alignment.

**Figure 6 microorganisms-10-01270-f006:**
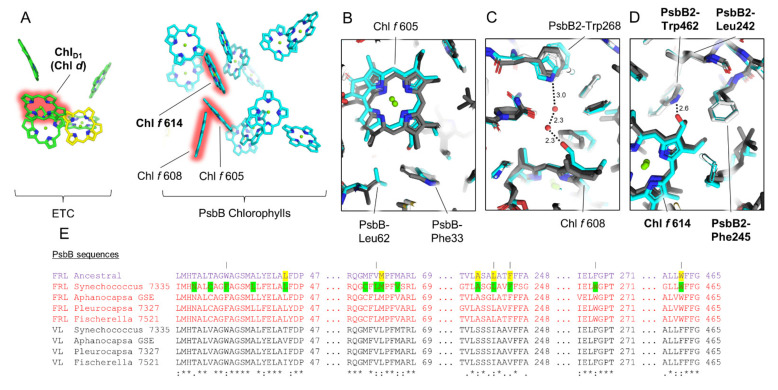
Conservation of FRL-PsbB interactions near the Chl *f* molecules bound to it. In (**A**), the Chls and pheophytins in the electron transfer chain and FRL-PsbB of the *Synechococcus* 7335 apo-FRL-PSII structure are shown as tetrapyrrole rings only from a membrane plane view. The Chls *d* and *f* are highlighted in red. In (**B**–**D**), the *Synechococcus* 7335 apo-FRL-PSII structure (cyan), the homology model of the FRL-PsbB ancestral sequence (white), and two non-FaRLiP holocomplex PSII structures (light and dark grey from *T. vulcanus* and *Synechocystis* 6803, respectively) are superimposed. (**B**–**D**) show the regions of the structures near the C2 formyl moiety of the three Chl *f* molecules bound to FRL-PsbB (sites 605, 608, and 614 based on the original assignment of Chl *a* sites from *T. vulcanus* PSII [46]), respectively. They also show H-bonding interactions involving the C2 formyl moieties of Chl *f* molecules in dashed lines with distances in units of Å where applicable. In (**A**–**D**), features that appear to be conserved in the ancestral FRL-PSII are labeled in bold font. In (**E**), a series of partial sequence alignments is shown that includes the FRL ancestral sequence and FRL- and VL-specific sequences from extant FaRLiP-capable cyanobacteria. Conserved FRL-specific residues in extant cyanobacteria are highlighted in green in the sequence from *Synechococcus* 7335. If the same position is conserved in the FRL ancestral sequence, it is highlighted in yellow. Vertical lines above residue positions in (**E**) correspond to amino acids labeled in (**B**–**D**). The Clustal Omega sequence conservation identifiers are shown below the alignment.

**Figure 7 microorganisms-10-01270-f007:**
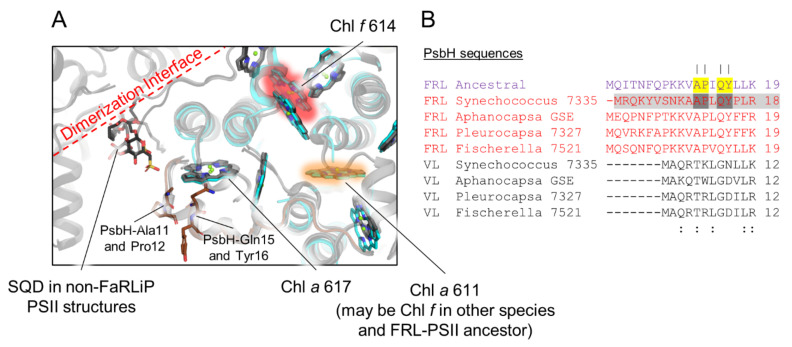
Conservation of FRL-PsbH residues. In (**A**), the *Synechococcus* 7335 apo-FRL-PSII structure (colored), the homology model of the FRL-PsbH ancestral sequence (white), and two non-FaRLiP holocomplex PSII structures (light and dark grey from *T. vulcanus* and *Synechocystis* 6803, respectively) are superimposed. Cartoons are shown with partial transparency and stick representations of applicable residues and cofactors are shown. For Chl molecules, only tetrapyrrole rings are shown. The Chl *f* assigned in PsbB of the *Synechococcus* 7335 apo-FRL-PSII structure is highlighted in red. The Chl *a* site that is suggested to bind Chl *f* in other species and the FRL-PSII ancestor is highlighted in orange. Additional views of the PsbH2 homology model can be found in Appendix A. In (**B**), a partial sequence alignment is shown that includes the FRL-ancestral sequence and FRL- and VL-specific sequences from extant FaRLiP-capable cyanobacteria. Note that the sequence from *Synechococcus* 7335 is highlighted in grey to signify that there are presently no corresponding structural data on this subunit. Conserved FRL-specific residues in extant cyanobacteria are highlighted in dark grey in the sequence from *Synechococcus* 7335. If the same position is conserved in the FRL ancestral sequence, it is highlighted in yellow. Vertical lines above residue positions in (**B**) correspond to amino acids labeled in (**A**). The Clustal Omega sequence conservation identifiers are shown below the alignment.

**Figure 8 microorganisms-10-01270-f008:**
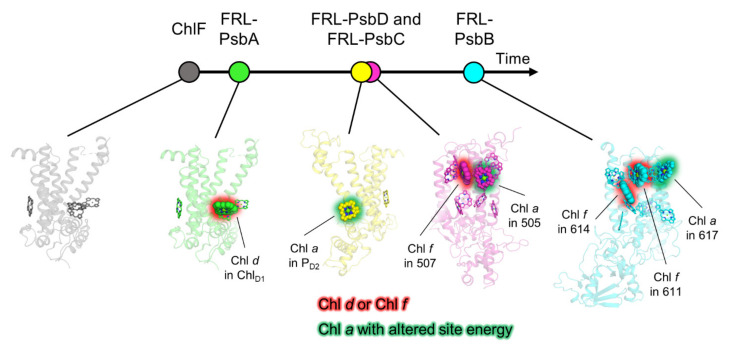
Proposed sequence of events during the emergence of FRL-PSII as described in the text.

**Table 1 microorganisms-10-01270-t001:** Comparison of conserved FRL-specific residues in FRL-PSII of *Synechococcus* 7335 and the ancestral sequences.

FRL Sequence from *Synechococcus* 7335	FRL-Specific Positions Conserved in the Ancestral Sequence (%)	FRL-Specific Positions Conserved Only in the Extant Sequences (%)
FRL-PsbA	3.3	3.3
FRL-PsbB	2.4	6.3
FRL-PsbC	3.9	3.9
FRL-PsbD	1.7	1.7
FRL-PsbH	6.1	7.6

## Data Availability

The phylogenetic data presented in this study are available in the
Appendix A.

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
