# Peer review of "Molecular Evolution of Far-Red Light-Acclimated Photosystem II"

_microorganisms, 2022, doi:10.3390/microorganisms10071270_

Round 1

Reviewer 1 Report

Review for the paper "Molecular evolution of far-red light-acclimated photosystem II" by Christopher J. Gisriel, Tanai Cardona, Donald A. Bryant and Gary W. Brudvig submitted to "Microorganisms".

General comment.

Cyanobacteria are unique among chlorophototrophs by virtue of their ability to photo-oxidize water and evolve oxygen, a reaction that has globally important consequences. Marine cyanobacteria are presently estimated to contribute more than 25% of primary production on Earth. Far-red light photoacclimation is a mechanism that allows some cyanobacteria to utilize far-red light for oxygenic photosynthesis. During Far-red light photoacclimation, cyanobacteria remodel photosystem (PS) I, PS II, and phycobilisomes while synthesizing Chl d, Chl f, and far-red-absorbing phycobiliproteins, and these changes enable these organisms to use FRL for growth. Photosystem II (PS) acts a water:plastoquinone oxidoreductase by producing a powerful oxidant (P680+) and a weak reductant (reduced plastoquinone, PQH2). PS II in cyanobacteria mostly occurs in thylakoids as dimers, in which each monomer comprises 20 polypeptide subunits, 35 Chl a, 11 β–carotenes, 2 plastoquinones, 2 hemes, 1 non-heme Fe atom, 4 Mn atoms, and 3–4 Ca atoms. Currently, there is a lack of data concerning some mechanisms of far-red light photoacclimation. The authors summarized previously published data and processed these. They provided ancestral sequence reconstruction and structure-based molecular evolutionary studies of the FRL-specific subunits of PSII. They showed that the origin of FRL-PSII required the accumulation of a relatively small number of amino acid changes. The authors' analysis suggests a minimal model for engineering far-red light absorbance into plant PSII for biotechnological applications. I think that this report expands our knowledge regarding far-red light-acclimated photosystem II in cyanobacteria and may be of great interest to scientists dealing with bioenergetic systems. The authors used an adequate dataset and methods to process the data. Statistical methods seem to be valid and correctly used. The main results are illustrated with relevant Figures. Discussion is focused on the main findings.

I can recommend this high-quality paper for publication after minor revisions.

Specific remarks.

As the authors used previously published material, not their original data, I think that "Review" is a more appropriate paper type, not "Article". If the authors agree, they should also update the paper with "Conclusion".

L 125. Consider replacing “each of these were” with “each of these was”

L 255. Consider replacing “are found” with “is found”

L 423. Consider replacing “are shown” with “is shown”

L 465. Consider replacing “stick representation” with “stick representations”

L 558. Consider replacing “process” with “processes”

L 573. Consider replacing “imply” with “implies”

Author Response

Reviewer 1

General comment. 

Cyanobacteria are unique among chlorophototrophs by virtue of their ability to photo-oxidize water and evolve oxygen, a reaction that has globally important consequences. Marine cyanobacteria are presently estimated to contribute more than 25% of primary production on Earth. Far-red light photoacclimation is a mechanism that allows some cyanobacteria to utilize far-red light for oxygenic photosynthesis. During Far-red light photoacclimation, cyanobacteria remodel photosystem (PS) I, PS II, and phycobilisomes while synthesizing Chl d, Chl f, and far-red-absorbing phycobiliproteins, and these changes enable these organisms to use FRL for growth. Photosystem II (PS) acts a water:plastoquinone oxidoreductase by producing a powerful oxidant (P680+) and a weak reductant (reduced plastoquinone, PQH2). PS II in cyanobacteria mostly occurs in thylakoids as dimers, in which each monomer comprises 20 polypeptide subunits, 35 Chl a, 11 β–carotenes, 2 plastoquinones, 2 hemes, 1 non-heme Fe atom, 4 Mn atoms, and 3–4 Ca atoms. Currently, there is a lack of data concerning some mechanisms of far-red light photoacclimation. The authors summarized previously published data and processed these. They provided ancestral sequence reconstruction and structure-based molecular evolutionary studies of the FRL-specific subunits of PSII. They showed that the origin of FRL-PSII required the accumulation of a relatively small number of amino acid changes. The authors' analysis suggests a minimal model for engineering far-red light absorbance into plant PSII for biotechnological applications. I think that this report expands our knowledge regarding far-red light-acclimated photosystem II in cyanobacteria and may be of great interest to scientists dealing with bioenergetic systems. The authors used an adequate dataset and methods to process the data. Statistical methods seem to be valid and correctly used. The main results are illustrated with relevant Figures. Discussion is focused on the main findings.

I can recommend this high-quality paper for publication after minor revisions.

Specific remarks.

As the authors used previously published material, not their original data, I think that "Review" is a more appropriate paper type, not "Article". If the authors agree, they should also update the paper with "Conclusion".

We will discuss this point with the editor for further insight, but at present, we prefer to maintain the “Article” paper type. This is because the following findings have not been reported previously and are not a review of the current literature: (1) the far-red light-specific residues are identified with much less noise than previous due to a larger sequence set, (2) ancestral sequences are calculated for the first time using up-to-date sequence datasets, (3) the far-red light-specific interactions of chlorophyll a molecules in the antenna are newly revealed, and (4) the homology model of PsbH2 reveals N-terminal conservation.

L 125. Consider replacing “each of these were” with “each of these was”

Change made.

L 255. Consider replacing “are found” with “is found”

Change made.

L 423. Consider replacing “are shown” with “is shown”

Change made.

L 465. Consider replacing “stick representation” with “stick representations”

Change made.

L 558. Consider replacing “process” with “processes”

Change made.

L 573. Consider replacing “imply” with “implies”

Change made.

Reviewer 2 Report

The manuscript titled "Molecular evolution of far-red light-acclimated photosystem II" describes the evolutionary reconstruction of several PSII subunits that are specialized in far-red light induced photosynthesis. The work looks at published sequence data to identify conserved residues that may be responsible for preferential Chl d or f binding. The work is very straightforward and directly addresses the goals with a descriptive focus on many individual amino acid changes and their potential impact on this evolutionary adaptation. As such, this work is speculative and predictive, with no direct measurements on the impact of the identified mutations.

Generally, the work would benefit from some trimming and simplification of the more complex descriptions of individual residues, as the narrative bogs down due the repetitive nature of addressing each PSII subunit separately. The work also surprisingly lacks context or comparison to the only known fully far-red photosystems in Acaryochloris.

Major comments:

A lack of comparison to the only known constitutive Chl d-binding PSII limits the scope of the manuscript's observations regarding conserved FRL-specific mutations. Some discussion of Acaryochloris photosystems and its place the the VL/FRL continuum would go a long way toward placing results in a greater context. In fact, Acaryochloris sequences only appear to be present in one tree (PsbC) where they are quite divergent and could be relevant in comparison to many of the conversations, even if these subunits share more homology to traditional VL photosystems.

How was the MRCA of cyanobacteria calculated? Given the wide variation in its depth within the tree and location related to FRL sequences, this estimation would seem to be a critical component to the discussion of this work's context.

Minor comments:

338 - Sentence starting with "It is of interest..." is not clear at all and likely needs to be edited for clarification.

All trees - The labels used for all trees are very raw and should be cleaned up to be more easily readable. Proper spacing and removal of the unnecessary and undescribed numbering (presumably only for data organization?) would greatly facilitate readability.

Fig 2/3 - The ribbon models are very small, blurry, and busy, making interpretation very difficult. Fig 2 could be increased in size wholesale, but Fig 3 could trim the grey-out subunit or split into two figures to highlight the complex elements more.

Supp Fig 1 - Why make a special notation for the branch that includes 7335 when you could simply label that branch using 7335 as the representative sequence?

Supp Fig 5 - The font in part A is too compressed and unreadable. Inclusion of panel B likely unnecessary and causing this compression.

Author Response

Reviewer 2

Comments and Suggestions for Authors

The manuscript titled "Molecular evolution of far-red light-acclimated photosystem II" describes the evolutionary reconstruction of several PSII subunits that are specialized in far-red light induced photosynthesis. The work looks at published sequence data to identify conserved residues that may be responsible for preferential Chl d or f binding. The work is very straightforward and directly addresses the goals with a descriptive focus on many individual amino acid changes and their potential impact on this evolutionary adaptation. As such, this work is speculative and predictive, with no direct measurements on the impact of the identified mutations.

Generally, the work would benefit from some trimming and simplification of the more complex descriptions of individual residues, as the narrative bogs down due the repetitive nature of addressing each PSII subunit separately. The work also surprisingly lacks context or comparison to the only known fully far-red photosystems in Acaryochloris.

Major comments:

A lack of comparison to the only known constitutive Chl d-binding PSII limits the scope of the manuscript's observations regarding conserved FRL-specific mutations. Some discussion of Acaryochloris photosystems and its place the the VL/FRL continuum would go a long way toward placing results in a greater context. In fact, Acaryochloris sequences only appear to be present in one tree (PsbC) where they are quite divergent and could be relevant in comparison to many of the conversations, even if these subunits share more homology to traditional VL photosystems.

Based on the reviewer’s comment, we have now added the following test to the results section: “We note that none of the trees showed evidence that the FRL sequences had any close relationship to those in the Chl d-producing Acaryochloris sp. strains. This observation supports the hypothesis that the facultative FaRLiP response is unrelated to the constitutive FRL-absorbance exhibited by Acaryochloris strains, and that the two strategies probably evolved convergently.”

And also in the Discussion: “While no evidence was found that the FRL sequences had any relationship to those used by Acaryochloris spp., there is still a possibility that the adaptations could be still related through the synthesis of Chl d.”

Because we find no evidence that that FaRLiP-capable cyanobacteria and Acaryochloris are related, we wish to avoid any greater comparison of the two. Beyond the description included in the text, we also note that Acaryochloris use Chl d for FRL-absorption whereas FaRLiP-capable cyanobacteria use Chl f. This third point is especially important because compared to FaRLiP-capable cyanobacteria that require site specificity for Chl d and f-binding among Chl a, Acaryochloris marina PSII subunits do not require site specificity for different Chl types except probably in the electron transfer chain where Chl a and Pheo a may bind. This further solidifies the profound differences between the two FRL-absorbing strategies.

We also note that an excellent pre-print was posted in biorxiv that makes valuable comparisons of the two PSII types from an energetic standpoint (https://www.biorxiv.org/content/10.1101/2022.04.05.486971v2.full). We now cite this pre-print in the Introduction.

How was the MRCA of cyanobacteria calculated? Given the wide variation in its depth within the tree and location related to FRL sequences, this estimation would seem to be a critical component to the discussion of this work's context.

The MRCA of cyanobacteria is not calculated, and it isn’t necessary for the calculation of ancestral states, which are calculated in an unrooted tree. In fact, the ancestral states are calculated for all occurring nodes in the unrooted tree, as the phylogeny itself is inferred. The ancestral states would be calculated mostly from the information available within the FRL sequences, and the neighboring nodes that would be VL sequences, but this process does not require knowledge of the rooting of the tree. We have updated the Materials and Methods to explicitly mention that these are calculated with unrooted trees and the Fig. 1 legend explains how the MRCA of cyanobacteria was defined.

In this instance we assigned the MRCA based on previous knowledge on the evolution of D1 subunits (e.g., Cardona et al. 2015; Sheridan et al. 2021) and on the evolution of cyanobacteria (e.g., Gloeobacterales branching as sister lineage to all other clades).

In the case of PsbB, PsbC, PsbD, and PsbH, in which the tree has been rooted at the divergence point of Gloeobacterales, there is no sequence being calculated for a node that would be equivalent to the MRCA of cyanobacteria, because there is no outgroup included that would identify it. In the figure, the tree has been visualized to represent Gloeobacterales as the earliest branch, creating an artificial node. We could visualize the trees as unrooted radial phylogram, but we thought this way would make the relative positions of the branch more apparent to the reader.

If we wished to calculate a sequence for the MRCA of cyanobacteria, we would need to add additional sequences to work as outgroup to create that node. For example, if we were to infer a tree of PsbC and PsbB together, since they are homologous, each subunit would work as outgroup for the other, so one could potentially acquire information on the ancestral sequences for both PsbC and PsbB, that one would predict could be also equivalent to that of the MRCA of cyanobacteria. This was done for PsbA and PsbD in the Oliver et al. 2021 BBA paper. It does come with caveats. Nonetheless, to retrieve the ancestral FRL sequences it is not necessary to have a rooted tree to infer an ancestral node for the MRCA of cyanobacteria.

The case for PsbA is a bit different because the phylogeny is more complex, in which there is a preserved history of PsbA duplications that antedate the MRCA of cyanobacteria. We based the visualization of the tree, on the previous work by Cardona et al. 2015.

Minor comments:

338 - Sentence starting with "It is of interest..." is not clear at all and likely needs to be edited for clarification.

We have altered this sentence: “Obtaining structural data of allophycocyanin in complex with PSII may aid in determining the roles of the individual residues at their interface that are conserved in FRL sequences.”

All trees - The labels used for all trees are very raw and should be cleaned up to be more easily readable. Proper spacing and removal of the unnecessary and undescribed numbering (presumably only for data organization?) would greatly facilitate readability.

We have provided the raw unedited data in the Supplementary Materials and follow a standard tree formatting, which can be visualized using any preferred tree visualizer. In both cases, each sequence has been assigned a unique number. The reason for these is that it makes it easier to search for each sequence, when inspecting the alignments and the trees, and when navigating the different paralogues that can be found within the datasets. We have also provided the accession number next to the species name, if the reader wishes to look for a specific sequence in a database (e.g. to find metadata on the genome), it is only a matter of using this number to search the NCBI.

Fig 2/3 - The ribbon models are very small, blurry, and busy, making interpretation very difficult. Fig 2 could be increased in size wholesale, but Fig 3 could trim the grey-out subunit or split into two figures to highlight the complex elements more.

We have increased the size of Fig. 2. As stated in the Fig. 3 legend, panel A is to show where the changes in PsbA2 are generally, and panel B zooms in on the specific residues. We are unsure what changes the reviewer is suggesting, but we feel that it serves the intended purpose in its current state.

Supp Fig 1 - Why make a special notation for the branch that includes 7335 when you could simply label that branch using 7335 as the representative sequence?

We collect and curate the data through a pipeline of bioinformatic tools. It turned out that a genome of a strain named cf. Phormidesmis sp. LEGE 11477 is a very close relative of Synechococcus 7335, and so in the 98% sequence identity curation, a couple of times the 7335 sequences did not make it as the former was kept instead. We thought it would be best to leave the data as unaltered as possible since the fact does not change the conclusions in any way. As it felt odd to just simply rename the branch. When the PsbA dataset was collected for the first time, these were not intended to be used for this specific project, but for other explorations of the reaction center proteins phylogeny. It is also somewhat inconvenient to redo the process, as this is very time consuming. I hope the reviewer understands. We thought it would be ok to just make the remark.

Supp Fig 5 - The font in part A is too compressed and unreadable. Inclusion of panel B likely unnecessary and causing this compression.

We have reformatted the figure to make it bigger and easier to read.